# Post-Wildfire Debris Flows in Montecito, California (USA): A Case Study and Empirically Based Debris Volume Estimation

**Diwakar KC and Liangbo Hu \***

Department of Civil and Environmental Engineering, University of Toledo, Toledo, OH 43560, USA; diwakar.kc@rockets.utoledo.edu
*   Correspondence: liangbo.hu@utoledo.edu

**Abstract:** Wildfires have a strong influence on various geotechnical and hydraulic properties of soils and sediments, which may become more vulnerable to landslides or debris flows. In the present study, a case investigation of the 2018 post-wildfire debris flows in Montecito, California, USA, was conducted, with a focus on the wildfire-affected areas and debris volume estimation. Significant debris were deposited around four major creeks, i.e., Montecito Creek, San Ysidro Creek, Buena Vista Creek, and Romero Creek in January, 2018, one month after the Thomas fire. Satellite images utilizing remote sensing techniques and geographic information system (GIS) data were analyzed to identify areas affected by the wildfire. Relevant data, including the slope, catchment area, and rainfall were used in two empirical models to estimate the debris volumes around the four creeks. As compared with field observation, each debris volume estimated with these empirical models was within the same order of magnitude. The debris volumes were generally underestimated when using the rainfall recorded at the Montecito Weather Station; the estimates considerably improved with the rainfall record from the Doulton Tunnel Station. The results showed that, overall, such empirical approaches are still of benefit for engineering practice, as they are capable of offering first-order approximations. The accuracy and availability of rainfall data are critical factors; the rainfall data in mountainous areas are generally higher than in the low lands, and consequently were more suitable for debris volume estimation in the present study, where the debris flows typically occurred in areas with steep slopes and at higher elevations.

**Keywords:** wildfire; remote sensing; geographic information system; rainfall; empirical models; debris flow



## 1. Introduction

Wildfires remain a persistent threat to many communities around the world. In the United States, an area of nearly 29,000 km$^2$ was burned in almost 59,000 wildfire events in 2021 [1]. In the state of California, USA, thousands of wildfires occur each year; between 2010 and 2020, there were at least 55 large wildfires, each of which blazed over more than 10 km$^2$ [2,3]. Wildfires can lead to considerable alterations in the physical, chemical, mechanical, and biological properties of burned soils and sediments [4–7], which consequently may become more susceptible to various gravity driven geological hazards, such as landslides, debris flows, mudslides, and floods. It has been widely reported that burned soils become more resistant to water infiltration, leading to enhanced runoff or overland flow [8,9], while being more vulnerable to erosion [10,11]; consequently, they have much greater potential to generate massive flows of debris material [12–14]. As a matter of fact, such post-wildfire hazards have become more widespread in the western states of the United States. The recent example of the Glenwood Canyon mudslides in Colorado after a season of intense wildfire led to the closure of Interstate 70 for several weeks in July 2021.

Post-wildfire hazard assessment plays an important role in mitigating the risk of such hazards [14–16]; in particular, the range, scope, and size of potential debris flows

are among the most important information in an assessment. In engineering practice, the estimation of the debris flow volume that may be generated from a burned watershed can be of great benefit for debris flow hazard assessment [17,18]. Although the physical and mechanical processes of debris flows involve complex interactions among potentially many phases of material constituents, which are in motion and have been under intensive investigation with various theoretical and numerical developments emerging over the last few decades [19–25], empirically based methods remain a useful tool in practice for predicting debris volume. Different empirical models have been developed, as early as 1980s, for the assessment of debris volume, employing several variables, such as rainfall, basin characteristics, and material type [26–29]. Later, such empirical models were further extended for the assessment of post-wildfire peak flows [30] and debris volume estimation [17,31–35] by incorporating wildfire-related variables. These methods are generally based on a multivariate logistic regression analysis, using past data or evidence to identify key variables and establish their relationships with the debris volume. They can be rapidly applied with popular geographic information systems (GIS) for the assessment of debris flows at real large-field scale.

In the present study, the well-known 2018 Montecito County (California) post-wildfire debris flows are examined. This area experienced a massive wildfire, which started on 4 December 2017 and continued for almost 40 days, before it was fully contained. One month after the wildfire, i.e., on 9 January 2018, massive debris flows occurred, which resulted in 23 casualties and over 100 people injured, more than 400 buildings damaged, and the closure of a major highway; the total economic loss was over $1 billion [36,37]. In this case study, the areas affected by the wildfire were identified and the rainfall records were analyzed to assess the role of rainfall in the subsequent debris flows. Two empirical models were employed to estimate the debris volumes in different parts of the study area, where the debris flows were concentrated. The results were finally compared with the relevant previously reported field observations. The present study could have important practical implications in this area. The Montecito area is of great interest for the assessment of potential future hazards, as the geological formations in the Santa Ynez Mountains in the north of Montecito are very fragile, and the potential for landslides and debris flows is high; such prospects may impact various settlements located on the southern side of the mountains. In addition, wildfires have become a recurring, almost seasonal, event in this area, and their effects on potential landslides or debris flows need to be better understood, in order to properly develop relevant mitigation strategies.

## 2. Study Area

The study area of the Montecito debris flows is located approximately 90 miles northwest of Los Angeles, in the vicinity of Montecito Peak in California, USA. It is in the Santa Barbara County of California; the Santa Ynez Mountains are in the north of the study area. These mountains are composed of weak sandstones and shale, which are especially susceptible to weathering and incisions [38]. Due to excessive weathering, these mountains have significant loose soil layers on the surface, which are often covered by vegetation. There are well-developed human settlements extending from the foot of these mountains to the coastline of the Pacific Ocean in the area, oriented east–west. Numerous small water channels flow dendritically, forming four major creeks, i.e., Montecito Creek, San Ysidro Creek, Buena Vista Creek, and Romero Creek, all of which travel through the human settlements. The lowest temperature in the area is around 6 °C during December and January, while the highest temperature is around 27 °C in September. The average annual rainfall in the Montecito area is about 486 mm. In general, the driest month in this area is July, with approximately 0.5 mm rainfall, and the wettest is February, with around 113 mm rainfall; however, large spatial variations in the rainfall distribution across this area and extreme rainfall events are common [37,39].

The wildfire-affected area is in the mountainous region. As shown in the slope map presented in Figure 1, the slope of most catchment areas in the mountains is more than 30%;

however, the area in the southern part of the mountains is less steep, and human settlements are common. The maximum elevations of the catchment areas are 962.0 m, 967.8 m, 512.1 m, and 895.6 m for Montecito Creek, San Ysidro Creek, Buena Vista Creek, and Romero Creek, respectively. The catchment areas of Montecito Creek, San Ysidro Creek, Buena Creek, and Romero Creek are 12.48 km$^2$, 7.84 km$^2$, 1.74 km$^2$, and 5.15 km$^2$, respectively.

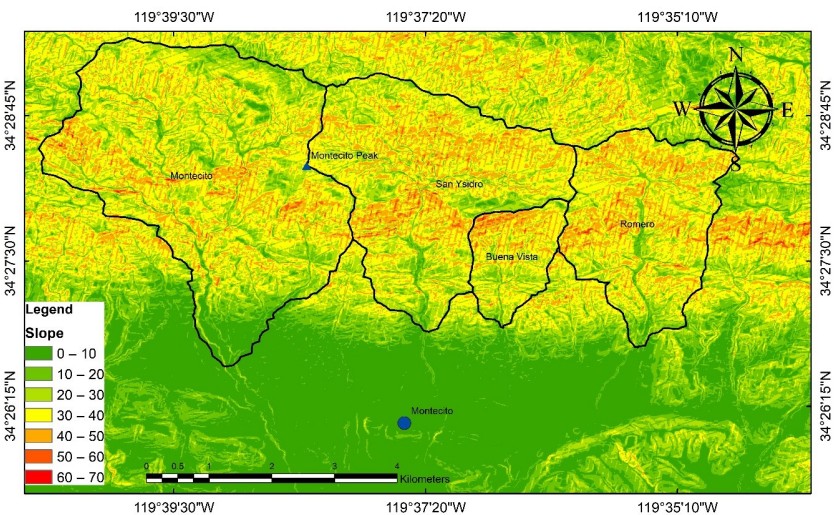

**Figure 1.** Slope map of the study area. The four polygons represent the catchment areas of the four creeks that experienced debris flow. The blue dot indicates the town of Montecito.

It should be noted that the steep mountains in this area were highly eroded after the rainfall and likely susceptible to debris flows. The sediment for the debris flows originated from the hill slopes, rather than from stream bed erosion. Both rill and inter-rill erosion were observed in the watershed. No major debris flows or landslides were initiated in the watershed area. It was observed that the debris flow was predominantly a viscous fluid, formed from a mixture of water and eroded fine soil. The bulked flow in the creeks transported the large boulders from upstream of the creeks (the northern part of the area) to the downstream (the middle and southern part of the area). Some bank erosion, due to the large flow in the upstream, also contributed boulders to the flow. Although the solid material for the flow came from different sources, the major initiation was supplied by the erosion of the watershed in the Santa Ynez Mountains [36,37,40]. It can be concluded that the bulk of the flow resulted from considerable erosion, which caused the subsequent debris flow.

## 3. The 2017 Wildfire and Subsequent 2018 Debris Flows

### 3.1. Wildfire-Affected Areas

A massive wildfire (later named as the Thomas fire) broke out from a line slap, i.e., collision between two power lines at a cattle ranch on Anlauf Canyon Road on 4 December 2017; it burned a total of 1140 km$^2$ in Ventura County and Santa Barbara County, including many parts of San Ynez Mountains in the northern part of Montecito. The wildfire destroyed 1063 structures and killed one civilian and one firefighter [41]. The wildfire continued to expand for forty days, before it was fully contained.

In the present study, the wildfire-affected areas were identified based on data acquired with remote sensing techniques. Remote sensing data are openly available from several different agencies; for example, the Landsat satellite images captured by the National Aeronautics and Space Administration (NASA) are provided by the United States Geological Survey [42], and the sentinel satellite images captured by the European Space Agency (ESA) are available from the Copernicus Open Access Hub [43].

In the present study, Landsat 8 data [42] acquired on 25 December 2017, after the fire, were used. The path, row, UTM Zone, and datum of the retrieved data were 042, 036, 11,

and WGS1984, respectively. Various RBG false color maps were generated to identify the fire-affected areas. For example, in Landsat 8 data, the combination of Band 7 short-wave infrared (SWIR), Band 5 near-infrared (NIR) and Band 2 (visible band) as RBG can be analyzed to successfully detect wildfire-affected areas based on the light absorption and reflection of plants. The photosynthetic pigments in plants absorb red and blue light, while reflecting green light; hence, the areas unaffected by wildfire appear greenish, as green light is reflected, while the red and the blue lights are absorbed by plants. Meanwhile, the wildfire-affected areas appear reddish after the plants that absorb red and blue light are burned. Figure 2 shows an RBG image of the entire study area, where the burned areas can be distinguished by a reddish color. The blue dot represents the location of the town of Montecito. In the northern part of Montecito are the Santa Ynez Mountains, where the Montecito Peak is indicated by a black triangle in Figure 3. The main area affected by the post-wildfire debris flows is enclosed by a yellow square in Figure 3 and was the focus of the present study.

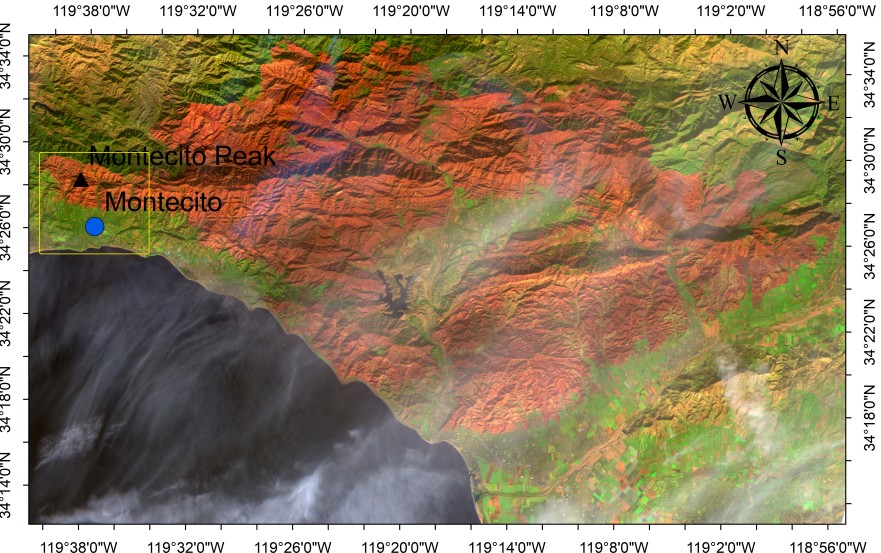

**Figure 2.** The fire affected areas appear reddish in the post-wildfire image, taken shortly after the burning destroyed the plants that typically absorb red and blue light; this color (RGB) image was generated using a combination of Band 7 (SWIR), Band 5 (NIR), and Band 2 (visible band).

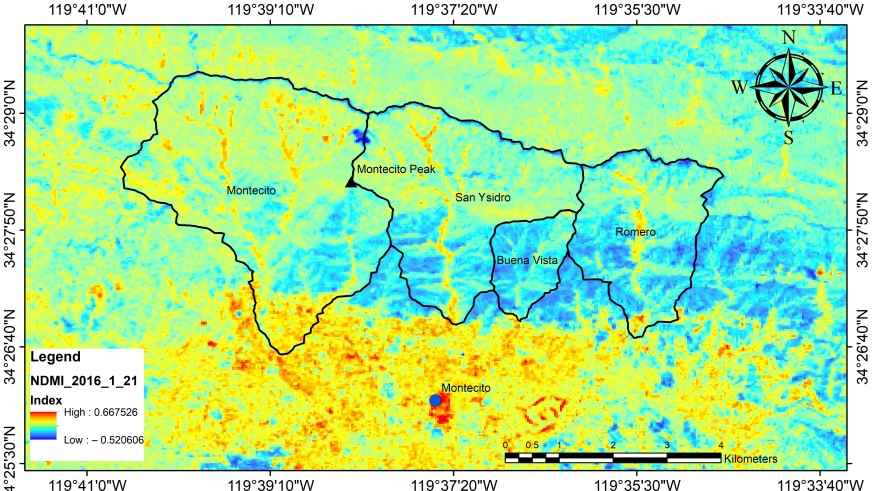

**Figure 3.** NDMI map based on data acquired on 21 January 2016. The four polygons from left to right are the catchment areas of Montecito creek, San Ysidro creek, Buena Vista creek, and Romero creek.

*3.2. Vegetation Burned and Its Regrowth*

The vegetation cover present in the area before the wildfire and immediately after the wildfire was analyzed to examine the effect of the fire and the range of its impact. In addition, the vegetation cover one month after wildfire was also studied, to examine the vegetation regrowth when the area had received sufficient rainfall. Therefore, overall, three Landsat satellite datasets were collected and analyzed; first, the Landsat 8 data acquired on 21 January 2016 were retrieved, to analyze the vegetation cover before the fire; second, the data acquired on 25 December 2017 represented the conditions immediately after the wildfire; third, the data acquired on 26 January 2018 reflected the short-term land cover and vegetation regrowth after the wildfire. The normalized difference moisture index (NDMI) was used to analyze the vegetation cover. The values of the NDMI range from −1 to +1; a low value indicates low vegetation coverage, and a high value high coverage. The NIR and SWIR band values were used for the NDMI calculation:

$$\text{NDMI} = \frac{\text{NIR} - \text{SWIR}}{\text{NIR} - \text{SWIR}} \tag{1}$$

The use of NIR and SWIR mitigates the effects of the atmosphere and illumination. SWIR is sensitive to the water content in vegetation and the mesophyll of leaves. The NIR band picks up the bright reflectance from the internal structure and dry matter content of a leaf; therefore, this combination offers a reasonably high accuracy for vegetation cover change [44].

Figure 3 presents the NDMI map based on the data acquired on 21 January 2016. It shows that there was considerable vegetation cover in the northern part of the catchment area of the Montecito Creek, San Yasidro Creek, and Romero Creek. The southern part of the basin, along with almost the entire catchment area of the Buena Vista Creek, had low vegetation cover. To the south of these catchments, there was dense vegetation cover, which appears yellowish and reddish in the map; this represents the widespread human settlements in this area. Evidently, the highest concentration of human settlements is located around the town of Montecito.

Figure 4 shows an NDMI map based on the data acquired on 25 December 2017. Overall, the NDMI values decreased significantly across the entire area; this suggests that the wildfire had consumed most of the vegetation. The lowest NDMI values are in blue color on the map, and this clearly shows that the vegetation in the mountainous areas had mostly been burned. Figure 5 shows the NDMI map one month after the wildfire, based on the data acquired on 26 January 2018. The mountainous areas that appear blue represent low NDMI values, these values increased somewhat from those immediately after the wildfire in Figure 5. Hence, there was certainly some evidence of vegetation growth; however, these burned areas still had a very low NDMI, and this suggests that the vegetation regrowth was fairly modest.

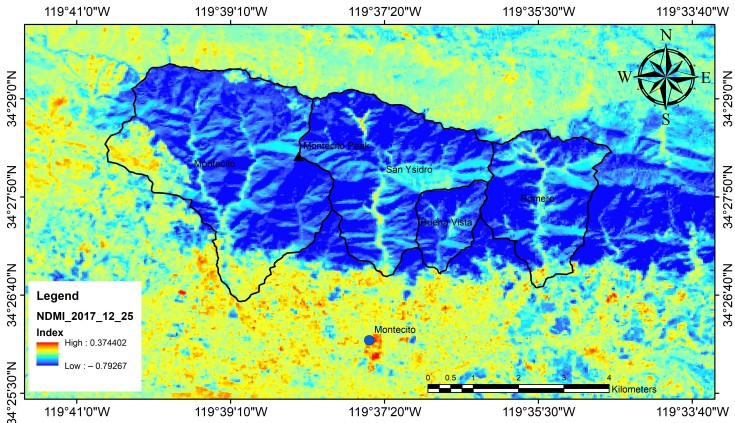

**Figure 4.** NDMI map based on the data acquired on 25 December 2017.

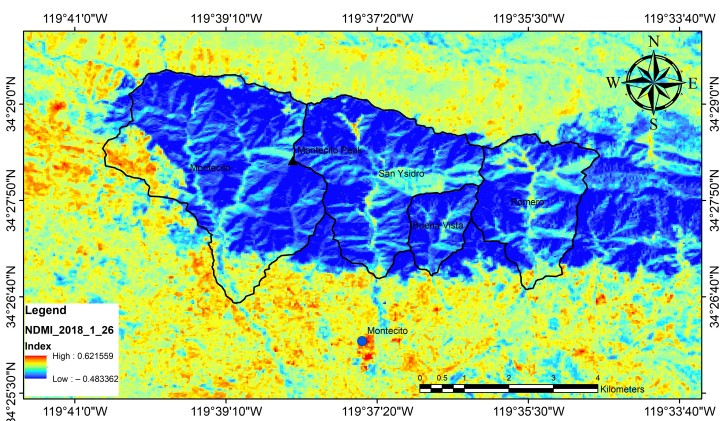

**Figure 5.** NDMI map based on the data acquired on 26 January 2018.

### 3.3. The 2018 Debris Flows and Rainfall

Debris flows are often triggered by intensive rainfall, and it is of interest to analyze the rainfall data in this region. In the present study, the rainfall data of this area, available from an online database, namely World Weather Online (hereafter referred to as WWO), provided by Zoomash Ltd., London, UK [45] were examined, focusing on the month of January for the years 2016, 2017, and 2018, and in particular, the day of 9 January 2018, when massive debris flows occurred. Starting from 12:00 am on January 1, the cumulative rainfall (mm) is plotted for each January in Figure 6.

The results show significantly more rainfall in the year 2017 than in 2018. The intensity of rainfall on 22 December 2017 was higher than that on 9 January 2018; at 9 am, 12 am, and 3 pm, the 3 h interval rainfall on the former date was 18.1 mm, 18.2 mm, and 18.3 mm, respectively, while on the latter date it was 4.2 mm, 4.4 mm, and 12.8 mm, respectively. It should be noted that, in 2018, an intense rainfall started only on Jan 8 and then further intensified after 20 h and was maintained for almost one and half days. Incidentally, this period coincided with the initiation of debris flows in the study area.

It is worth noting that the explored database does not offer comprehensive coverage of rainfall variations across this region. There were significant spatial variations of rainfall, along with extremely high intensities, across the study area [37,39]. In the present study, the rainfall data recorded by the National Oceanic and Atmospheric Administration [39] were also explored. Table 1 summarizes the total 2-day rainfall on 8 January and 9 January 2018, collected at different rain gauge stations in the study area.

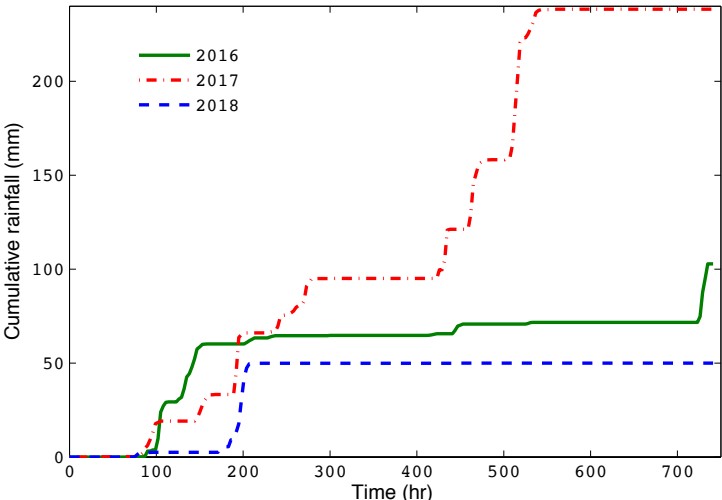

**Figure 6.** Rainfall in the month of January of 2016, 2017, and 2018, provided by the WWO database [45].

Table 1 clearly reveals large spatial variations of rainfall across this region. The locations of these rainfall stations are shown in the elevation map in Figure 7. These results indicate that the rainfall was higher in the mountains (KYDT, Doulton Tunnel) than in the lower lands along the coastline (Summerland, Montecito, Santa Barbara) in this region. The debris flows were likely initiated by the high intensity rainfall that occurred on the steep mountain slopes. In particular, the maximum 1 h rainfalls recorded at the surrounding rainfall stations, Doulton Tunnel, KYDT, and Montecito, were 39 mm, 28 mm, and 24 mm, respectively, which was most likely the triggering factor for the debris flows [37]. It is of interest to compare the rainfall intensity recorded in the study area with the rainfall threshold proposed for the initiation of debris flows in the existing literature [14,46]. Calculated based on the model developed by Staley et al. [14], the 15 min, 30 min, and 1 h rainfall thresholds in the study area would be 31, 20, and 12 mm/h, respectively. Similarly, these rainfall thresholds would be 56, 30, and 16 mm/h, respectively, based on the model proposed by Berti et al. [46]. At the Montecito Station, the recorded 15 min, 30 min, and 1 h rainfall intensities were 74, 39, and 24 mm/h, respectively, while at the Doulton Tunnel Station, these three intensities were 104, 63, and 39 mm/h, respectively; they were considerably higher than the threshold values and, indeed, suggest strong possibilities of debris flows. Hence, the rainfall occurring at two rainfall stations (the highest rainfall from the mountainous area, i.e., Doulton Tunnel, and the lowest rainfall from the lower land, i.e., Montecito) was used to estimate the range of debris volume in the subsequent analysis.

**Table 1.** Total rainfall on 8 January and 9 January 2018, at the different gauge stations [39].

| Station | Elevation (m) | Rainfall (mm) | Latitude | Longitude |
|---------|---------------|---------------|----------|-----------|
| KTYD | 724 | 81 | 34°28′16″ | 119°40′37″ |
| Doulton Tunnel | 541 | 91 | 34°27′28″ | 119°33′52″ |
| Montecito | 41 | 54 | 34°25′39″ | 119°38′25″ |
| Summerland | 26 | 56 | 34°24′55″ | 119°34′53″ |
| Santa Barbara | 17 | 54 | 34°25′12″ | 119°42′00″ |

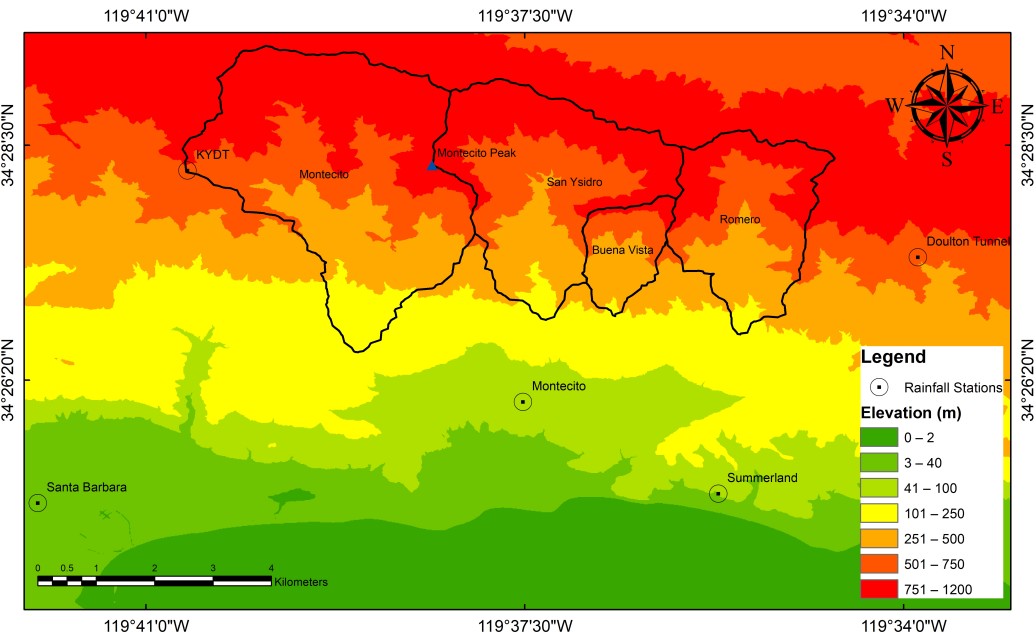

**Figure 7.** Location of the different rainfall stations in the elevation map.

## 4. Debris Volume Estimation

### 4.1. Mathematical Models

Various models have been developed for the assessment of post-wildfire debris volume, based on a number of factors, including storm frequencies, peak discharge, erosion rates, fire intensity, vegetation index, relief ratio, and so on [17,33,34,47–50]. Table 2 summarizes some of the prominent models for post-wildfire debris volume estimation. In general, the applicability of each model is largely dependent on a specific locality, because the relevant model parameters were established based on the data from a certain range of regions or areas. Two models are of particular benefit for the area of interest in the present study. Cannon et al. [33] developed a model based on data from the Intermountain West region of the United States, i.e., Colorado, Utah, Montana, Idaho, and southern California. The model was established using multiple-liner-regression analysis of burn extent, soil properties, basin morphology, and rainfall. Gartner et al. [34] developed an empirical model by considering sediment volume, burn severity distribution, watershed morphology, the time duration since the fire, rainfall storm conditions, and soil properties in Ventura, Los Angeles, and San Bernardino Counties of southern California. This model was developed for two debris volume estimates: the debris volume within two years after the wildfire in the short term, and the debris volume after two years, i.e., in the long term. The former was of particular interest to the present study, as the debris flows occurred in just a few months after the wildfire in Montecito. Both models are suitable for cases in southern California and thus were used in the present study. One of the major benefits is that the number of parameters required is not excessively high, and they can be readily calculated or generated in GIS. It is also of interest to explore both models, since the parameters considered in the two models are quite different, as detailed below. One of the most important parameters is rainfall; the former model considers the total rainfall in a storm, but the latter considers 15 min peak rainfall intensity.

The first estimate was based on the equation proposed by Cannon et al. [33], as follows (hereafter referred to as Model 1),

$$\ln V = 7.2 + 0.6 \ln A + 0.7 B^{1/2} + 0.2 T^{1/2} + 0.3 \tag{2}$$

where $V$ is the volume of debris generated (in $m^3$), $A$ is the catchment area with a slope greater than 30% (in $km^2$), $B$ is the area burned with high or moderate severity (in $km^2$), and $T$ is the total rainfall (in mm).

The second estimate for the debris volume, $V$, was made based on the empirical equation developed by Gartner et al. [34] in a short term, i.e., within two years of the fire (hereafter referred to as Model 2),

$$\ln V = 4.22 + 0.39 i_{15}^{1/2} + 0.36 \ln B + 0.13 R^{1/2} \tag{3}$$

where $i_{15}$ is the 15 min time period peak rainfall intensity (in mm/h) and $R$ is the relief, which is the difference between the highest elevation of the catchment and the lowest elevation of the catchment (in m). It is noted that the same parameter $B$, the watershed area burned at high or moderate intensity (in $km^2$), as used in Equation (2), was also employed here in this estimation. The area of the catchment and the relief could be readily determined from geographic information systems (GIS) using the digital elevation data (DEM) of the study area.

### 4.2. Input Parameters

As discussed in Section 3, the rainfall data varied spatially, and the two stations located at Montecito and Santa Ynez Mountains were representative of the low land and the high mountains, respectively, and consequently both were explored in the present study. The maximum 15 min rainfall intensities recorded by NOAA on 9 January 2018, at the Montecito Station and at the Doulton Tunnel Station were 74.16 mm/h and 104.68 mm/h, respectively. The total rainfall recorded in an hour at those two stations (Montecito and Doulton Tunnel)

was 24 mm and 39 mm, respectively. The rainfall parameters at these two different locations were used to estimate the debris volume. Although the other parameters remained the same, for the sake of clarity, the input parameters for the debris volume estimation are summarized in Table 3 using the rainfall data collected at the Montecito Station, while those parameters considering the rainfall at the Doulton Tunnel Station are listed in Table 4.

**Table 2.** Summary of certain post-wildfire debris volume estimation models.

| Source | Parameters Considered | Study Area |
|---|---|---|
| Gartner et al. [17] | Total storm rainfall, burned area, particle size distribution rainfall intensity, catchment area with slopes ≥30% | Western US (Colorado, Utah, California) |
| Cannon et al. [33] | Total storm rainfall, burned area, catchment area with slopes ≥30% | Colorado, Montana, Idaho, Southern California |
| Gartner et al. [34] | Peak 15 min rainfall intensity, burned area, relief ratio | Southern California |
| Rowe et al. [47] | Storm frequencies, peak discharge, erosion rates, fire correction factors, vegetation index, relief ratio | Southern California |
| Gatwood et al. [48] | Peak 1 h precipitation, peak discharge, fire factor, catchment area, relief ratio | Southern California |
| Pak and Lee [49] | Peak 1 h rainfall intensity, total storm rainfall, fire factor, time since burn, number of events causing erosion, relief ratio, catchment area | Southern California |
| Gartner et al. [50] | Peak 1 h rainfall intensity, burned area, average gradient, time since most recent fire, catchment area, relief ratio | Southern California |

**Table 3.** Input parameters for the debris volume estimation based on the rainfall data at the Montecito Station.

| Creek | $A$ (km$^2$) | $B$ (km$^2$) | $T$ (mm) | $i_{15}$ (mm) | $R$ (m) |
|---|---|---|---|---|---|
| Montecito | 10.62 | 11.5 | 24.13 | 74.16 | 716 |
| San Ysidro | 7.30 | 7.84 | 24.13 | 74.16 | 727 |
| Buena Vista | 1.64 | 1.74 | 24.13 | 74.16 | 277 |
| Romero | 4.76 | 5.15 | 24.13 | 74.16 | 627 |

**Table 4.** Input parameters for the debris volume estimation based on the rainfall data at the Doulton Tunnel Station.

| Creek | $A$ (km$^2$) | $B$ (km$^2$) | $T$ (mm) | $i_{15}$ (mm) | $R$ (m) |
|---|---|---|---|---|---|
| Montecito | 10.62 | 11.5 | 39.16 | 104.64 | 716 |
| San Ysidro | 7.30 | 7.84 | 39.16 | 104.64 | 727 |
| Buena Vista | 1.64 | 1.74 | 39.16 | 104.64 | 277 |
| Romero | 4.76 | 5.15 | 39.16 | 104.64 | 627 |

*4.3. Results*

Table 5 summarizes the debris volume estimations with the two different methods using the rainfall recorded at the Montecito Station, along with the volume observed in the field [37] while Table 6 presents the results using the rainfall recorded at the Doulton Tunnel Station. It is worth noting that the volume observed from the field is based on the estimates of the area affected and the depth of debris deposition observed in the field; they serve as preliminary estimates and are not intended to be precise results.

**Table 5.** Estimated debris volumes using the rainfall data recorded at the Montecito Station.

| Creek | Observed (m$^3$) | Model 1 (m$^3$) | Difference (%) | Model 2 (m$^3$) | Difference (%) |
|---|---|---|---|---|---|
| Montecito | 231,000 | 213,996 | −7.36 | 152,717 | −33.89 |
| San Ysidro | 297,000 | 113,012 | −61.95 | 136,638 | −54.00 |
| Buena Vista | 41,000 | 16,363 | −60.08 | 20,780 | −49.32 |
| Romero | 100,000 | 60,283 | −39.72 | 91,446 | −8.55 |

**Table 6.** Estimated debris volumes using the rainfall data recorded at the Doulton Tunnel Station.

| Creek | Observed (m$^3$) | Model 1 (m$^3$) | Difference (%) | Model 2 (m$^3$) | Difference (%) |
|---|---|---|---|---|---|
| Montecito | 231,000 | 280,080 | +21.25 | 287,226 | +24.34 |
| San Ysidro | 297,000 | 147,910 | −50.20 | 256,987 | −13.47 |
| Buena Vista | 41,000 | 21,417 | −47.76 | 39,082 | −4.68 |
| Romero | 100,000 | 78,898 | −21.20 | 171,991 | +71.99 |

Considering the rainfall recorded at the Montecito Station (Table 5), the volume estimated with Model 1 for the Montecito Creek was very close to the field observation, but for all other creeks the estimated volume was more than 39% lower; in particular, the volumes for the San Ysidro Creek and the Buena Vista Creek were greatly underestimated, by over 60%. Model 2 also underestimated the volume of debris; however, the differences were more moderate, with the highest underestimation of 54% for the San Ysidro Creek and the lowest of only 8.55% for the Romero Creek.

The results in Table 6, produced by considering the rainfall recorded at the Doulton Tunnel Station in the Santa Ynez Mountains, overall seem better matched with the field observations, except for the estimation for the Romero Creek with the second method. The volume for the Montecito Creek was overestimated by around 20% with both models, but overall there were more underestimations. Overall, Model 2 produced results fairly consistent with the field observations, even though the overestimation for the Romero Creek was high percentage wise, the sheer difference in the volume was actually not excessively high. As a matter of fact, except for the San Ysidro Creek, no differences were greater than 100,000 m$^3$, and even in this case, Model 2 using the rainfall data from the Doulton Tunnel Station was able to generate a fairly close estimation. Using the rainfall data from the Montecito Station seemed to underestimate the debris volumes, possibly because the rainfall collected in the low land was lower than for most of the area. This suggests that the rainfall data collected in mountainous areas, such as the Doulton Tunnel Station, could be more suitable for such estimations. Whereas the first model relied on only the overall rainfall, the second model used the 15 min peak rainfall intensity, which produced higher estimations and seemed to be more accurate for the case examined in the present study. Overall, all predictions were within the same order of magnitude as the debris volumes observed in the field; the benefits of such empirical approaches could be still be of interest for engineering practice, as they are capable of offering first-order approximations.

## 5. Conclusions

The present study focused on the 2018 debris flows in the Montecito area after the Thomas fire; debris was deposited around four major creeks: Montecito Creek, San Ysidro Creek, Buena Vista Creek, and Romero Creek. The wildfire-affected areas could be identified with data collected using remote sensing technologies. The rainfall was analyzed and shown to be a major trigger for the ensuing debris flows. The volumes of debris material were estimated with two empirical models. However, the rainfall varied considerably across the study area, and hence data from two weather stations were used in the present study. Two empirical models were employed to estimate the debris volumes deposited in the burned watersheds using the actual topography of the study area in GIS. Compared

with the reported estimations based on the field observations, the results from these empirical models were of mixed success. The model utilizing a short-duration rainfall intensity [34] seemed to produce better estimations overall, while the model using the overall rainfall [33] generally underestimated the volumes in the present study. The accuracy and availability of rainfall data also played an important role; the rainfall in mountainous area was generally higher than in the low lands, and thus could have been more suitable for debris volume estimation in the present study, as debris flows typically occur in areas with steep slopes at higher elevations. Overall, the practical benefits of such empirical approaches remain significant, as they are capable of offering first-order approximations. It is worth noting that such typical empirical approaches are entirely built on data or evidence from past events. Fundamental processes of infiltration, erosion, and debris motion involving soil properties or watershed characteristics [51] need to be studied to adequately address the influences of wildfires on soils and sediments and better understand the critical mechanisms involved in their increased vulnerability to debris flows.

**Author Contributions:** Conceptualization, D.K.; methodology, D.K. and L.H.; software, D.K.; formal analysis, D.K.; investigation, D.K.; resources, L.H.; data curation, D.K.; writing—original draft preparation, D.K. and L.H.; writing—review and editing, L.H.; visualization, D.K.; supervision, L.H.; project administration, L.H. All authors have read and agreed to the published version of the manuscript.

**Funding:** This research received no external funding.

**Data Availability Statement:** Data used in the present study is contained within the article.

**Acknowledgments:** The authors wish to acknowledge the financial support provided by the University of Toledo through a Summer Research Fellowship during the preparation of this manuscript.

**Conflicts of Interest:** The authors declare no conflict of interest.

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
