# Peer review of "Post-Wildfire Debris Flows in Montecito, California (USA): A Case Study and Empirically Based Debris Volume Estimation"

_geotechnics, doi:10.3390/geotechnics3020020_

Round 1

Reviewer 1 Report

This manuscript focuses on the post wildfire debris flow, which is an important issue in engineering domain for disaster risk assessment. The data obtained from remote sensing and GIS were used to identify the areas affected by the wildfire and estimate the debris volumes. The findings are interesting and useful. I would like to suggest to accept this manuscript after a minor revision. Some suggestions are listed below.

(1) Abstract: Some key findings also should be included in abstract.

(2) Lines 18~20: It is better adding some references to support the source of the used data.

(3) Line 33: Debris flow?

(4) Lines 41~47: It is suggested to add a table to summarize the existing empirical models, and the comparisons between these models are also useful.

(5) Lines 64~66: Actually, this issue was not addressed in the current manuscript. I suggest to add it into the discussion part.

(6) Lines 50~66: The questions you want to answer in the manuscript are not clear. It seems that the testing of the suitability of the empirical models and the assessment of the potential causes or triggers for subsequent debris flows are the main scopes. Please make them more explicit.

(7) Have you made any improvements for the empirical models?

(8) Figure 1 is useless. You can consider to remove it.

(9) Lines 78~83: The figures presenting the annual data of rainfall and temperature would be much better.

(10) Lines 96~97: This sentence is odd.

(11) Conclusion part: It is more useful to list the key points of the findings one by one. In addition, some expressions are not appropriate, e.g., “the second model”, because the readers would be confused. I suggest using some words that would give the features of the empirical models.

Author Response

We are grateful to both reviewers for their constructive comments. The manuscript has been revised accordingly.

It is noted that in the submitted revised manuscript, only the parts related to major changes are highlighted in red color; in order to avoid potentially excessive distraction, minor changes, including individual sentence revision, minor details addition, wording improvement, typographical corrections, are kept in regular font and color within the text.

This manuscript focuses on the post wildfire debris flow, which is an important issue in engineering domain for disaster risk assessment. The data obtained from remote sensing and GIS were used to identify the areas affected by the wildfire and estimate the debris volumes. The findings are interesting and useful. I would like to suggest to accept this manuscript after a minor revision. Some suggestions are listed below.

(1) Abstract: Some key findings also should be included in abstract.

Reply: we have slightly modified with some additional findings in the revised manuscript. Obviously the key results of the debris volume estimation are of very mixed success and it is difficult to offer a general description without elaboration on the overall context (this is done in the Results and Conclusion sections), especially in the Abstract. Some general and specific findings are included in the revised abstract.

(2) Lines 18~20: It is better adding some references to support the source of the used data.

Reply: indeed, the ref [1] provided by the National Centers for Environmental Information (NCEI) is added.

(3) Line 33: Debris flow?

Reply: the term “debris” was used to refer to the debris material in the original manuscript, in the revised manuscript we have changed it to “debris flow” to avoid potential confusion.

(4) Lines 41~47: It is suggested to add a table to summarize the existing empirical models, and the comparisons between these models are also useful.

Reply: we thank the reviewer for this suggestion. A table summarizing some of the prominent existing empirical models is included in the revised manuscript (Table 2), where the used parameters used in each model are summarized.

(5) Lines 64~66: Actually, this issue was not addressed in the current manuscript. I suggest to add it into the discussion part.

Reply: Indeed, this comment is offered to present a context and motivation on the present study, obviously this issue is not the core part, and beyond the scope of the present investigation. We have added a brief statement (see Comment #6) to avoid the potential confusion.

(6) Lines 50~66: The questions you want to answer in the manuscript are not clear. It seems that the testing of the suitability of the empirical models and the assessment of the potential causes or triggers for subsequent debris flows are the main scopes. Please make them more explicit.

Reply: the questions in Line 50~60 of the original manuscript is the focus of the present study, and in our opinion, is addressed in Section 3 and 4 where the areas affected by the wildfire are identified and the rainfall records are analyzed, as well as the debris volume estimations are investigated. We have slightly revised these statements for clarification. The statements in Line 60~66 are implications of the study and intended to highlight the importance and benefit of the study, but not the technical focus of the investigation (See Comment #5). We have added the brief statement “The present study could have strong practical implications in this area” to avoid potential confusion.

(7) Have you made any improvements for the empirical models?

Reply: We must admit that we did not make attempts to improve the empirical models. Our main focus beyond the scope of the present study is extended to investigate the fundamental mechanisms involved in the processes of infiltration, erosion and debris motion that are affected by the wildfires and lead to enhanced vulnerability to debris flows, as mentioned in the Conclusion, we added a recent reference ([51], by the first author) in the revised manuscript so that interested readers may explore his further efforts.

(8) Figure 1 is useless. You can consider to remove it.

Reply: we have removed it in the revised manuscript.

(9) Lines 78~83: The figures presenting the annual data of rainfall and temperature would be much better.

Reply: The details on the rainfall associated with debris flows are already given in section 3.3. The annual rainfall and temperatures are not the major concern of the study, as the present study is more concerned with estimating the debris volume generated from individual rainfall events rather than annual rainfall. Therefore, they are just mentioned in the text as part of background information.

(10) Lines 96~97: This sentence is odd.

Reply: this sentence originally describes the field observation of the constituents of the debris (fine solid + water), and large solids (boulders) were carried by the flow and deposited in the downstream. This obviously is common in debris flow but not necessarily trivia, as it will have strong implications in the numerical modeling of debris flow, for example, if a multi-phase (solid, fine solid and fluid) model is used. The second reviewer also raised some concern about this statement, we have revised this statement to avoid the confusion.

(11) Conclusion part: It is more useful to list the key points of the findings one by one. In addition, some expressions are not appropriate, e.g., “the second model”, because the readers would be confused. I suggest using some words that would give the features of the empirical models.

Reply: It is probably a matter of taste whether to use bullet list or just plain text to summarize the findings. We find that most of the articles in this journal adopt the traditional text option. In our opinion, it is of no essential significance and we have left it unchanged (unless the editor mandates the format revision). We do agree that the expressions about “the first model” or “the second model” could be vague, we have replaced them with the key feature and cited ref, such as “The model utilizing a short-duration rainfall intensity [34].....”.

Reviewer 2 Report

Lines 34-35

“In engineering practices the volume of debris that may be generated from the burned watershed can be of great benefit for the debris flow hazard assessment[ 16,17].“ Perhaps the estimation of the debris-flow volume. This of benefit in all the watersheds of the world.

Line 40  delete “geotechnical” it also concerns geomorphology, geology and hydraulics engineering. In addition add the references Medina et al. (2008) and Gregoretti et al. (2019)

Line 44  add the reference Marchi et al. (2019)

Line 97-98 “Large boulders were deposited in the depositional basin and subsequently transported from the upstream to the downstream along with the concentrated flow.” Meaningless sentence

Line 99 “Some bank erosion due to large flow in the upstream”  upstream of what?  Unclear sentence.

Lines 199  Please compare the fallen rainfall with the rainfall threshold of Staley et al. (2017) and Berti et al. (2020)

Please use also the relationship of Marchi et al. (2019) for the sediment volume estimation

low-

References

Berti M., Bernard M., Gregoretti C., Simoni A. (2020) Physical interpretation of rainfall thresholds for runoff-generated debris flows. Journal of Geophysical Research: Earth Surface, 125, doi: 10.1029/2019JF005513

Marchi, L., Brunetti, M.T., Cavalli, M., Crema, S. 2019. Debris-flow volumes in Northeastern Italy: relationship with drainage and size probability.  Earth Surface and Landforms, 44: 933-943.

Staley, D. M., Negri, J. A., Kean, J. W., Laber, J. L., Tillery, A. C., & Youberg, A. M. (2017). Prediction of spatially explicit rainfall intensity duration thresholds for post-fire debris-flow generation in the western United States. Geomorphology, 278, 149–162. https://doi.org/10.1016/j.geomorph.2016.10.019

Some sentence is unclear

Author Response

We are grateful to both reviewers for their constructive comments. The manuscript has been revised accordingly.

It is noted that in the submitted revised manuscript, only the parts related to major changes are highlighted in red color; in order to avoid potentially excessive distraction, minor changes, including individual sentence revision, minor details addition, wording improvement, typographical corrections, are kept in regular font and color within the text.

Lines 34-35, "In engineering practices the volume of debris that may be generated from the burned watershed can be of great benefit for the debris flow hazard assessment[ 16,17].“ Perhaps the estimation of the debris-flow volume. This of benefit in all the watersheds of the world.

Reply: yes, agreed, we have revised the sentence to address the estimation of the debris-flow volume.

Line 40  delete “geotechnical” it also concerns geomorphology, geology and hydraulics engineering. In addition add the references Medina et al. (2008) and Gregoretti et al. (2019)

Reply: “geotechnical” has been delete. The two references ([23], [25]) have been added.

Line 44  add the reference Marchi et al. (2019)

Reply: this ref [29] has been added.

Line 97-98 “Large boulders were deposited in the depositional basin and subsequently transported from the upstream to the downstream along with the concentrated flow.” Meaningless sentence

Reply: this sentence originally describes the field observation of the constituents of the debris (fine solid + water), and large solids (boulders) were carried by the flow and deposited in the downstream. This obviously is common in debris flow but not necessarily trivia, as it will have strong implications in the numerical modeling of debris flow, for example, if a multi-phase (solid, fine solid and fluid) model is used. We have revised this statement to avoid the confusion.

Line 99 “Some bank erosion due to large flow in the upstream”  upstream of what?  Unclear sentence.

Reply: “upstream” means the upstream of the creeks (northern part of the area), which is mentioned in the revised manuscript (see the last comment).

Lines 199  Please compare the fallen rainfall with the rainfall threshold of Staley et al. (2017) and Berti et al. (2020)

Reply: We are grateful to the reviewer to direct our attention to these two references (one of them was already cited in the original manuscript). Indeed, based on the models proposed by these two studies, our calculations show that the rainfall intensities in the study area considerably exceeded the threshold value suggested by these two studies and confirmed the strong possibilities for debris flows. A detailed discussion is added in Line 203~213.

Please use also the relationship of Marchi et al. (2019) for the sediment volume estimation.

Reply: One of the key features associated with each model is the area or region where the investigation is located and where the model is mainly applicable. The explore models in the present study are developed based on the cases in the Southern California of USA and intended for applications in this region. The suggested additional model was mainly developed for Northeastern Italy and therefore it can be argued that it may not be suitable for the present study, from the topography and geology perspective.

More importantly, the models explored in the present study are primarily aimed at the post wildfire cases; for example, the fire factor or burned area is a key parameter in these models; whereas the suggested additional model is more of a general debris estimation model without special attention to wildfires, hence the results are probably not a fair comparison. Therefore we decide not to pursue further in the present study (nonetheless this reference [29] is included and interested readers may explore it in future investigations). We hope that the reviewer would agree with our choice.